# Robotic Removal and Collection of Screws in Collaborative Disassembly of End-of-Life Electric Vehicle Batteries

**DOI:** 10.3390/biomimetics10080553

**Published:** 2025-08-21

**Authors:** Muyao Tan, Jun Huang, Xingqiang Jiang, Yilin Fang, Quan Liu, Duc Pham

**Affiliations:** 1School of Information Engineering, Wuhan University of Technology, Wuhan 430070, China; muyaotan@whut.edu.cn (M.T.); fangspirit@whut.edu.cn (Y.F.); quanliu@whut.edu.cn (Q.L.); 2School of Mechanical and Electronic Engineering, Wuhan University of Technology, Wuhan 430070, China; xingqiangjiang@whut.edu.cn; 3Department of Mechanical Engineering, School of Engineering, University of Birmingham, Birmingham B15 2TT, UK; d.t.pham@bham.ac.uk

**Keywords:** human–robot collaboration, disassembly, robotic unfastening, screws, electric vehicle batteries, end-of-life electric vehicles

## Abstract

The recycling and remanufacturing of end-of-life (EoL) electric vehicle (EV) batteries are urgent challenges for a circular economy. Disassembly is crucial for handling EoL EV batteries due to their inherent uncertainties and instability. The human–robot collaborative disassembly of EV batteries as a semi-automated approach has been investigated and implemented to increase flexibility and productivity. Unscrewing is one of the primary operations in EV battery disassembly. This paper presents a new method for the robotic unfastening and collecting of screws, increasing disassembly efficiency and freeing human operators from dangerous, tedious, and repetitive work. The design inspiration for this method originated from how human operators unfasten and grasp screws when disassembling objects with an electric tool, along with the fusion of multimodal perception, such as vision and touch. A robotic disassembly system for screws is introduced, which involves a collaborative robot, an electric spindle, a screw collection device, a 3D camera, a si*x*-axis force/torque sensor, and other components. The process of robotic unfastening and collecting screws is proposed by using position and force control. Experiments were carried out to validate the proposed method. The results demonstrate that the screws in EV batteries can be automatically identified, located, unfastened, and removed, indicating potential for the proposed method in the disassembly of EoL EV batteries.

## 1. Introduction

Due to environmental protection and resource conservation, the disassembly of end-of-life (EoL) electric vehicle (EV) batteries has drawn extensive attention as a key step in recycling and remanufacturing for a circular economy. Given the inefficiency, high costs, potential health risks, and various challenges associated with traditional manual disassembly, robotic disassembly and human–robot collaborative disassembly technologies have been increasingly developed and employed [1,2]. Robotic disassembly has been extensively researched and applied across multiple domains. In electronic product disassembly, cognitive robotic technology was developed for processing various product models [3]. This approach significantly enhanced the flexibility and adaptability of the disassembly process. For the disassembly of small fasteners, a vision-based robotic strategy for hex screw removal was proposed that incorporated camera pose adjustment, screw pose calculation, and coordinated motion planning between the robot and tools to achieve highly efficient and precise screw disassembly [4]. It provided an effective solution for handling the large number of screw removal tasks in electronic products.

Studies on developing automated disassembly processes for EV battery recycling have been conducted. For instance, a hybrid disassembly framework that utilises modified robotic arms and specialised tools was developed to optimise parameters such as design, safety, and cost [5]. This approach significantly reduced the overall disassembly steps while improving efficiency. Additionally, a knowledge-driven, flexible human–robot hybrid disassembly line was designed by implementing hierarchical task decomposition and intelligent planning [6]. This method addressed screw disassembly tasks in complex scenarios, providing innovative solutions for the disassembly of EV batteries. Robotic disassembly technology encompasses multiple key technologies. A flexible screw head detection system was developed by using YOLOv5, which classified six types for robot tool adaptation, achieving high precision and runtime efficiency for automated remanufacturing disassembly [7]. Tactile and force sensing was applied in operations such as screw insertion, ensuring accuracy and safety with a contact force feedback [8]. A force-sensing-based Markov decision process model was developed to adjust operations in real time during screw insertion based on force feedback, thereby improving success rates and stability [9]. Additionally, a spiral search method based on force and torque sensor data was investigated to enable robots to locate screws and adjust tool–screw engagement autonomously. The four distinct stages of screw removal, further optimising disassembly efficiency, were analysed [10].

Human–robot collaboration (HRC) plays a crucial role in the flexible disassembly of EoL products. A literature review of the HRC in non-destructive disassembly was conducted within industrial environments, examining collaborative principles, key elements, and existing challenges [11]. Theoretical foundations were provided for optimising human–robot collaborative models. Furthermore, an experimental HRC disassembly cell was developed, which involved robots and operators working cooperatively in a shared workspace [12]. This system could handle the uncertainties in complex disassembly tasks while enhancing both flexibility and operational efficiency. A multi-robot task collaborative allocation method was proposed by combining the K-Means algorithm with the particle swarm algorithm to leverage synergy [13].

In HRC disassembly, screws can be initially unfastened by a robot. Then, the loosened screws are removed and collected by a human operator [2]. Most automated screw disassembly technologies focus on unfastening or unscrewing operations, lacking a systematic solution for the robotic disassembly and collection of screws using vision, force, and position controls. In this paper, a novel method for the robotic unfastening, removal, and collection of screws is proposed to enhance the efficiency of EV battery disassembly. It learns from how human operators disassemble screws. Using multimodal perception such as vision and touch, human operators can loosen screws with an electric tool and grasp and remove them with their fingers. Therefore, machine vision and force sensors are employed for the multimodal perception of robots with an electric spindle, achieving high precision in unfastening. A collection device was developed to collect screws and drop them into a box, functioning like human fingers. Additionally, the proposed method was designed and optimised by analysing human screw disassembly operations to improve its efficiency, adaptability, and robustness.

The remainder of this paper is organised as follows. Section 2 provides a brief literature review, covering the HRC disassembly of EV batteries and the automated disassembly of screws. Section 3 describes a robotic disassembly system built in the authors’ laboratory for unfastening and collecting screws. Section 4 reports on the process for robotic unfastening and collecting screws. To validate the proposed method, disassembly experiments using screws were carried out on an EoL EV battery. This is reported in Section 5. Section 6 concludes this paper and provides suggestions for future work.

## 2. Related Work

### 2.1. Human–Robot Collaborative Disassembly of EV Batteries

With the rapid development of the EV industry, the recycling and remanufacturing of EoL EV batteries have become the focus of sustainable manufacturing and the circular economy. The manual disassembly of EV batteries is labour-intensive, low-efficiency, and high-cost, and it poses risks in disassembly operations [14]. Automated disassembly using robots and equipment lacks flexibility, making it challenging to handle the uncertainties and unpredictability of disassembly processes [15]. Human–robot collaboration provides a flexible, semi-automated approach for disassembling EV batteries [16]. Human–robot collaboration disassembly (HRCD) combines the high levels of accuracy, speed, and repeatability of industrial robots with the superior flexibility, dexterity, and cognitive capabilities of human operators [17], thereby mitigating the effects of uncertainties and potential dangers.

Human–robot collaboration overcomes the disassembly problems of EV batteries with partial automation by incorporating sensor-integrated robotics in more fields of human activity [18]. The implementation of human–robot collaboration for disassembling lithium-ion batteries was presented [19]. In HRC, human operators performed complex tasks, while robots were used to conduct repetitive and tedious tasks, such as removing screws and bolts. The automation potential of EV battery disassembly was assessed, and the technical feasibility, process challenges, and optimisation pathways were analysed [20]. Evaluation showed that most unscrewing operations should be automated, and most of the remaining operations should be performed by human operators.

In terms of HRC sequence optimisation and safety, an HRC flexible system was proposed to address the uncertainties in disassembly processes for EV battery recycling [21]. The architecture of the disassembly system was designed by analysing uncertainties, including features, connection methods, and destruction structures. An HRC disassembly task sequence optimisation method was investigated using multi-agent deep reinforcement learning with the QMIX architecture [22]. A multi-agent reinforcement learning environment with partial observability was designed based on a specific EVB structure. An HRC disassembly workstation was demonstrated to verify the QMIX-HRC disassembly strategy. A Large Language Model (LLM) with tailored prompts was employed to increase the efficiency of HRCD sequence planning [23]. The feasible disassembly sequences generated by a Dirichlet Bayesian network were quantitatively analysed using the LLM.

### 2.2. Automated Disassembly of Screws

The efficiency of the disassembly of EV batteries can be significantly improved by automating the disassembly of screw connections, which account for approximately 40% of disassembly operations [24]. Scholars have developed tools and systems for the automated disassembly of screws in EoL products, promoting the development of a circular economy [25]. A screwdriver was attached to an electric motor for unscrewing, and a potentiometer was used for measuring the torques in unscrewing processes [26]. The monitored torque signals were used to assess the condition of unscrewing operations, and a diagnostic procedure was developed to detect conditions and make informed decisions. An automatic tool was designed for unscrewing various types of screws, which included those with damaged heads [27]. Slots on the screw head could be created by the tool, becoming new active surfaces where torque could be transmitted. A passive and compliant pneumatic torque actuator with vision sensing was designed, which incorporated a pneumatic power unit, a square shaft transmission limit module, a rod limit module, and a vision module. The concept of screw removal using a developed separation unit was realised for the automated disassembly of screws in EV batteries [28].

Additionally, various automatic disassembly methods of screws have been proposed and investigated to handle the uncertainties and variability in the physical condition of screws. A system combining force and visual sensing was proposed for the automated unfastening of screws in notebook computers to achieve accurate and efficient screw removal [29]. A system for screw detection and tool recommendation was developed for robotic disassembly using machine vision [30]. The object detection algorithm of screws was investigated to improve the accuracy of screw detection. A robotic screw disassembly system using multi-head tools was designed for the destructive and non-destructive disassembly of screws [31]. Force/torque sensors were employed to guide the force controls of robots. A robotic end-effector was designed for screwing and unscrewing bolts to improve operational flexibility [32]. A machine learning-based method was proposed to detect the state of a screwdriver using its torque data and to unfasten screw connections [33].

A multifunctional screw disassembly workstation was developed by incorporating an automated sleeve replacement device [34]. A screw-type recognition method based on attributes was proposed to determine the most suitable disassembly methods. Search methods for the disassembly of screws were employed to compensate for position errors and ensure the accurate engagement of disassembly tools with screws. A flexible screw connection-oriented disassembly method using HRC was proposed [2]. The bolts could be engaged and loosened automatically by a robot with tools and then collected and classified by a human operator. In a robotic disassembly platform for a plug-in hybrid electric vehicle (PHEV) battery, a magnet was used to pick up the loose screw by a robot with a nutrunner and then move and drop it into a collection box.

The above research mainly focused on the methods and tools for the automated unfastening of screws. Few studies have been conducted on the whole robotic disassembly process of screws, which includes unfastening, removing, and collecting screws. This paper presents a novel method for the robotic unfastening and collection of screws in EV battery disassembly, thereby increasing disassembly efficiency and freeing human operators from tedious and repetitive tasks.

## 3. A Robotic Unfastening and Collecting System of Screws

### 3.1. A Human–Robot Collaborative Disassembly Cell

Figure 1 shows the layout of a human–robot collaborative disassembly cell of EoL EV batteries developed in the authors’ laboratory. The cell includes a collaborative robot (Doosan Robotics H2515), a 3D vision system (Hikvision MV-DB500S-A), an electric spindle (Desoutter EME38-20J), a proprietary collection device, a batch-head changing device, a si*x*-axis force/torque sensor (ATI F/T Delta), an EoL EV battery, and other components. The collaborative robot is a 6-axis robot arm with a 25 kg payload, a reach of 1500 mm, and a repeatability of ±0.1 mm. In the vision system, an RGB-D camera is installed on the robot to capture image information for recognising and locating screws in the EV battery. The electric spindle, equipped with an adapter installed on the robot, is used to unfasten the screw automatically. The self-developed collection device is used to clamp the unfastened screw, then move and drop it into a collection box. Using the batch-head changing device, various types of adapters could be fitted to the electric spindle. The si*x*-axis force/torque sensor is mounted between the robot and the electric spindle to monitor the variations in forces and torques during the unscrewing process.

### 3.2. Vision System for Recognition and Location of Screws

The core of the vision system is an RGB-D intelligent stereo camera with a working distance of 500 mm to 1000 mm, a frame rate of 30 fps, and a maximum resolution of 1408 × 1024. The vision system employs active binocular stereo imaging technology, combining it with high-frame-rate RGB images and depth maps to construct high-precision three-dimensional spatial information around objects. YOLOv8 is used as the recognition algorithm in this paper, utilising a deeper network structure and more efficient feature extraction methods to enhance detection accuracy. Techniques such as multi-scale prediction and attention mechanisms were introduced to improve the model’s generalisation ability and reduce false detection.

About 2500 images of screws in EV batteries were collected under different lighting and background conditions. The size of each image was 1408 × 1024 pixels. To enhance the model’s generalisation ability and the dataset’s diversity, data augmentation techniques, including cropping, stitching, mirror transformation, greyscale conversion, brightness adjustment, and noise addition, were applied to process the original images, thereby preventing the model from overfitting. After screening, 2000 images were retained and divided into a training set (1600 images) and a validation set (400 images) with a ratio of 4:1. The LabelImg image annotation tool was used to label both the training set and the validation set accurately. Figure 2 shows the image after recognition. The type, position, and depth information of the screws in an EV battery could be obtained using a vision system.

### 3.3. Screw Unfastening and Collecting System

With the type and position information of a screw obtained by the machine vision system, the robot, equipped with an electric spindle, will select a suitable adapter using the batch-head changing device and then move and unfasten the screw automatically. This electric spindle’s controller drives the spindle and screwdriver to rotate, allowing for the setting of disassembly force, rotation direction, and time. The electric spindle is connected to the sleeve rod by a spring inside, which provides flexibility for disassembly operations.

As shown in Figure 3a, a collection device developed in-house is employed to clamp the unfastened screw, then move and drop it into a collection box. Figure 3b illustrates the idea that whether the collection device with two fingers can successfully clamp the unfastened screw depends on its geometric position relationship with the screwdriver and the screw. As there is a spring inside the electric spindle and the clamping finger, the screw cutter has a specific initial geometric relationship, resulting in clamping uncertainties. The particular relationship between them needs to be clarified to determine the geometric and mechanical conditions of successful clamping, as shown in Figure 3c. In the approach and engagement phase, the position relationship between the two fingers and the screwdriver is shown in Figure 3a. The coordinate system is defined. The axis of the screwdriver is the *Z*-axis, and the clamping plane is parallel to the X-Y plane.

The geometric relationships between the clamping fingers and the electric spindle fitted with an adapter change during the ‘open’ state and the ‘closed’ state, which affects the clamping success rate. The geometric conditions can be described as follows:(1)$$\left\{\begin{array}{c} L_{0}=D_{0}+L\\ L_{Z}=D_{0}^{\prime }+L+\delta \\ D_{0}^{\prime }=D_{0}-\Delta x \end{array}\right.$$
where $L_{0}$ is the distance between the A plane (located on the electric spindle) and the lower surface of the adapter, $D_{0}$ is the initial distance between the A plane and the upper surface of the adapter, L is the length of the adapter, $L_{Z}$ is the distance between the A plane and the upper surface of the clamp finger (the plane of the clamping screw), $D_{0}^{\prime }$ is the distance between the A plane and the upper surface of the adapter, δ is the height of the screw head, and Δx is the amount of expansion and contraction of the spring inside the electric spindle. The geometric relation can be expressed as follows:(2)$$L_{Z}<L_{0}$$

Substituting Equation (1) into Equation (2), we obtain the following:(3)Δx>δ

For the spring, the mechanical condition can be expressed as:(4)$$F_{s}=K\Delta x$$
where $F_{s}$ is the force of the deformed spring, and K is the elastic coefficient of the spring. For the clamping process, force analysis is conducted with the electric spindle, adapter, clamping finger, and screw as a unit. The electric spindle exerts a force, $F_{sd}$, and the internal spring exerts a force, $F_{s}$, on the screw. The screw gives it a reaction force N. There is a balance condition that can be expressed as follows:(5)$$F_{sd}=F_{s}+\mu \cdot N$$
where $F_{sd}$ is the force exerted by the electric spindle on the adapter in the *Z*-axis direction, μ is the coefficient of friction of the contact surface, and N is the reaction force of the screw. The clamping non-slip condition can be expressed as follows:(6)$$N\geq F_{l}=\frac{\tau_{max}\cdot R_{sc}}{\mu }$$
where $F_{l}$ is the lateral force, $\tau_{max}$ is the maximum torque, and $R_{sc}$ is the radius of the screw. Substituting Equations (4) and (5) into Equation (6) yields the following:(7)$$\Delta x\leq \frac{{F_{sd}-\tau }_{max}\cdot R_{sc}}{KL}$$

According to the geometric conditions in Equation (3) and the mechanical conditions in Equation (7), the following can be obtained:(8)$$\delta <\Delta x\leq \frac{{F_{sd}-\tau }_{max}\cdot R_{sc}}{KL}$$

As shown in Figure 3c, the clamping finger has a V-shaped notch with an angle of α, which can be set according to the types and sizes of screws to be removed. In this paper, α is 120°. To avoid the collision between the clamping finger and the screw or the adapter before clamping, the following requirement should be satisfied:(9)$$D_{g}>R_{sd}$$
where $D_{g}$ is the vertical distance between the end face of a clamping finger and the centre of the screw in its initial state. In the clamping working state (closed state) of the collection device, the following condition should be met:(10)$$D_{g}^{\prime }<R_{sc}$$
where $R_{sc}$ is the radius of the screw head. The above Equations (8)–(10) should be satisfied to ensure that the clamping fingers can successfully collect the unfastened screw.

## 4. The Process of Robotic Unfastening and Collecting a Screw

### 4.1. Disassembly Process

The screws in EV batteries could be automatically recognised, located, loosened, and collected using a robot equipped with machine vision, an electric spindle, a batch-head changing device, and a collection device. Figure 4 illustrates the developed process for robotic unfastening and collecting a screw, comprising five stages: identify and locate, select or change, approach and engage, unfasten, and collect and drop.

Identify and locate. The first step is to identify the type and find the positions of the screws in the EV battery using the vision system. The robot, equipped with a 3D camera, moves to the positions above the screws to capture images and depth information. The type and position information of screws can be obtained using machine learning (ML) algorithms, which will then be sent to the robot.Select or change. Based on the type of screws, the robot with an electric spindle moves to select or change a suitable adapter by using a batch-head changing device. Therefore, the robotic cell could disassemble various types of screws, improving its flexibility.Approach and engage. The robot, equipped with an electric spindle and adapter, moves and approaches the position above a screw to be disassembled. The electric spindle rotates in the direction of the screw fastening and searches for, then engages, the screw head. The fastening torque information regarding the electric spindle is employed to assess whether the adapter engages the screw head successfully.Unfasten. Once the adapter and screw head have engaged, the electric spindle rotates in the direction of unfastening. The screw will move out of the screw hole.Collect and drop. As the screw runs out enough distance from the screw hole, the two fingers of the collection device clamp the screw and follow its movement until the screw is free from the threaded hole. Then, the unfastened screw will be moved away and dropped into a collection box. The process is now complete.

### 4.2. Recognising and Locating Screws Based on Machine Vision

A flowchart of the recognition and localisation of screws based on machine vision is shown in Figure 5. Firstly, the robot, equipped with a 3D camera, moves to a position above the screws in an EV battery. Then, the robot sends a signal to the vision system to trigger the camera to capture images of and depth information on the screws. The TCP/IP protocol is employed between the robot and the vision system. ML algorithms like YOLOv8 are used to analyse the images and obtain the type and plane coordinates (x and y) of the screw. The depth coordinate (z) could be obtained by using the depth information. Finally, the type and 3D coordinate information of the screw are sent to the robot.

### 4.3. Clamping and Dropping Screws Using a Self-Developed Collection Device

Figure 6 illustrates the process of clamping and dropping the screws using a self-developed collection device. The collection device has a two-finger gripper. In its initial state, the collection device is open during the electric spindle searches and engages with the screw head, as shown in Figure 6a. The bottom surface is 2–5 mm away from the bottom surface of the screwdriver in the positive direction of the *Z*-axis (the bottom surface of the clamping device is higher than the screwdriver), avoiding interference and collision. After the screw exits the threaded hole with sufficient clearance (Figure 6b), the collection device clamps the screw and tracks its movement. Once the screw is completely out of the threaded hole, the robot with the collection device moves and drops the unfastened screw into a collection box, as shown in Figure 6d.

## 5. Experimental Results and Discussion

### 5.1. End-of-Life EV Battery

To validate the proposed method for unfastening and moving screws, experiments were conducted on the disassembly of an EoL battery from a hybrid EV. Figure 7 shows a photograph of an EoL EV battery. The battery mainly consists of screws, an upper housing part, a dehydration box, a BMS, battery modules, a lower housing part, metal copper bars, a battery filter, copper bar connectors, metal bars, a fuse, and so on. There are 74 screws with a size of M5 on the upper part of the housing, which can be disassembled using the human–robot collaborative disassembly cell in Figure 1.

### 5.2. Experimental Parameter Settings

During the disassembly process, interaction forces exist between the electric spindle and the robot, the electric spindle and the screw, and the collection device and the screw. Due to the robot’s rigid movement, which may cause damage to parts, a force and position mixing control is implemented. In the ‘identify and locate’ stage, the stiffness in all directions is high. In the ‘select or change’ stage, the electric spindle is in contact with the adapter, and the force in the *Z*-axis direction needs to be maintained. The stiffness in the Z direction is high, and the stiffnesses in the X and Y directions are low. In the ‘approach and engage’ stage, the adapter comes into contact with the screw head, and the stiffness in the *Z*-axis direction should be high for accurate positioning. In the ‘unfasten’ stage, the electric spindle follows the movement of the screw with low stiffness in all directions. In the ‘collect and drop’ phase, the robot has no contact with the screw, and the stiffness is high in all directions. The stiffness parameters of the robot at different stages of unscrewing are listed in Table 1.

The parameters of the electric spindle during the different stages of unfastening and collecting the screw are listed in Table 2, which includes rotation speed, minimum torque, target torque, and maximum torque.

### 5.3. Experimental Results

Based on the above settings, an experiment was designed to investigate the disassembly of screws. Figure 8 presents the images taken of the entire process of unfastening and collecting screws from the battery top cover. Figure 8a shows that the robot moved to the photo-taking position above the battery, where the images and depth information of the screws were captured using the camera in the vision system. Based on the type information of the screw, the robot moved the electric spindle to select a suitable adapter, as shown in Figure 8b. Then, the robot approached the position of the screw to be dismantled, as shown in Figure 8c. Figure 8d shows that the adapter engaged with the screw head. The screw is shown unfastening in Figure 8e, and the screw moved out of the threaded hole. Figure 8f shows that the collection device clamped the screw, and Figure 8g shows that the screw was completely moved out of the threaded hole. The unfastened screw was moved to a position above the collection box, as shown in Figure 8h, and dropped into the box when the gripper of the collection device was opened (Figure 8i).

The screws were successfully unfastened in 97 out of 100 tests. Among these, the screws were collected and dropped into the collection box 96 times, achieving a successful rate of 98.97%. The main reason for the failure to unfasten the screws was the positioning errors of the system. Cumulative location errors existed in the robot’s base coordinate system, the end-effector coordinate system, and the vision system coordinate system, resulting in errors in the identification and localisation of the screws. Additionally, in the three unfastening failure cases, the titled screws were not parallel to the *Z*-axis and became stuck in the threaded hole. Other uncertain physical conditions of the screw, such as corrosion, contaminants, and mechanical damage, may lead to unfastening failures. The reason for the failure to collect a screw is that the gripping conditions were not met due to unfastening failure, resulting in the inability to collect and move the unfastened screw.

### 5.4. Analysis of Force and Torque in Unfastening and Collecting Screws

Figure 9 shows the 6-DOF (degree of freedom) dynamic forces and torques measured by the si*x*-axis force/torque sensor installed between the robot and the electric spindle during the unfastening and collecting processes. Figure 9a,b display the 3-DOF forces and 3-DOF torques, respectively. It can be seen that the approaching and engaging stage occurs from 1.9 s to 3.9 s. During the robot’s approach to the screw position, the 3-DOF forces fluctuate between −3 N and 5 N, and the 3-DOF torque fluctuates between −0.5 Nm and 0.5 Nm. The robot applied a force in the -Z direction, and the electric spindle did not rotate. Then, the electric spindle rotated in the direction of tightening the screw at a speed of 43.6 rpm, searched in a spiral motion, and engaged with the screw head. The magnitude of Fz is around 10 N, and the torque Ty fluctuates around 1 Nm. The unfastening stage spans from 3.9 s to 5.2 s, during which the electric spindle rotates in the unfastening direction at a speed of 436 rpm. The magnitudes of Fz and Ty are around 25 N and −2.5 Nm, respectively. The collecting and dropping stage occurred between 5.2 s and 8.1 s.

## 6. Conclusions and Future Work

Disassembly is a critical step in recycling and remanufacturing EoL EV batteries. The human–robot collaborative disassembly of EV batteries as a semi-automated approach has been investigated and implemented to increase flexibility and productivity. The disassembly of screws is one of the most common disassembly operations, accounting for approximately 40% of all disassembly operations for EoL products. This paper presents a novel method for the robotic unfastening and collection of screws during the disassembly of EV batteries, aiming to increase disassembly efficiency and free human operators from dangerous, tedious, and repetitive tasks. A robotic system for unfastening and collecting screws was introduced. The disassembly process of a screw was divided into five stages: identify and locate, select or change, approach and engage, unfasten, and collect and drop. Experiments were conducted to validate the proposed method using an EoL EV battery. The main contributions of this work are as follows:

(1) A robotic system for unfastening and collecting screws was proposed. The screws could be automatically recognised, located, loosened, and collected using a robot equipped with machine vision, an electric spindle, a batch-head changing device, and a collection device.

(2) A 3D vision system was employed to recognise and locate the screws to obtain their type and position information. Based on the type of screws, the robot equipped with an electric spindle can select or change a suitable adapter for disassembling the screws.

(3) A collection device was developed for collecting screws and dropping them into a collection box. Once the screw reached the desired distance from the threaded hole, the collection device clamped the screw. After the screw was completely unfastened and moved out of the threaded hole, it was moved and dropped into a collection box.

(4) Experiments were conducted, and the results show that the screw was unfastened and collected successfully 97 times out of 100, yielding a successful rate of 97%. The self-developed collection device achieved a success rate of 98.97%.

Future work will focus on addressing the positioning problems of the vision system to enhance screw recognition and location accuracy. The structure of the collection device will be improved to increase its reliability and flexibility. Additionally, experiments will be conducted to address the potential failure modes of the proposed method, thereby enhancing its robustness and efficiency. Comparisons with other screw disassembly methods of screws will be carried out.

## Figures and Tables

**Figure 1 biomimetics-10-00553-f001:**
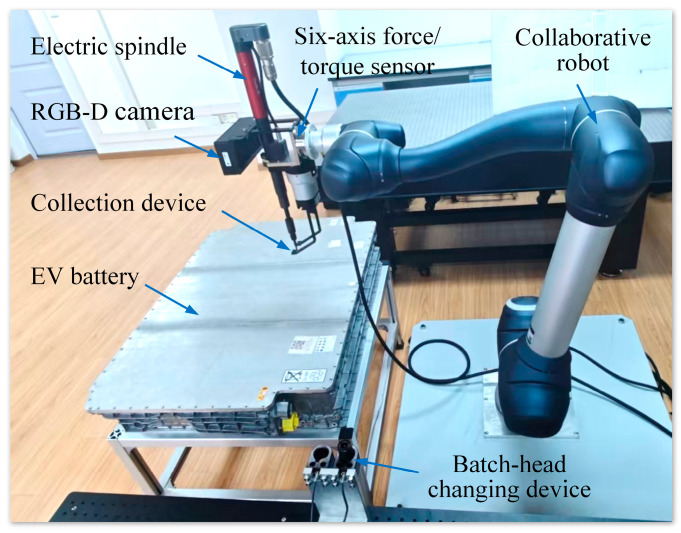
Layout of human–robot collaborative disassembly cell.

**Figure 2 biomimetics-10-00553-f002:**
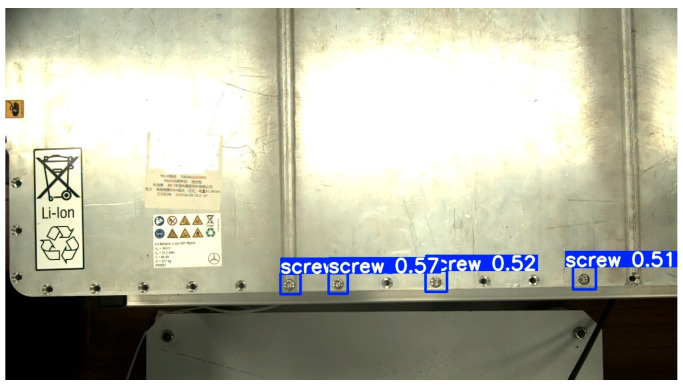
The image after recognition by the YOLOv8 algorithm.

**Figure 3 biomimetics-10-00553-f003:**
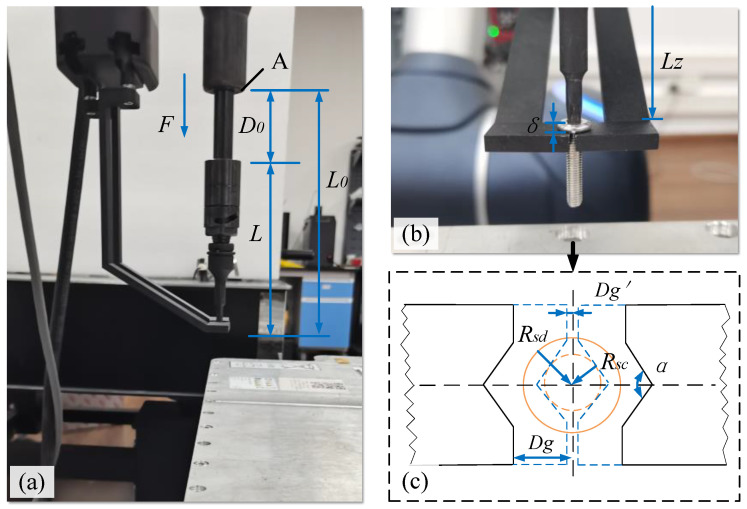
Positional relationships between the collection device, the electric spindle, and the screw. (**a**) The Y-Z plane, (**b**) X-Z plane, and (**c**) X-Y plane.

**Figure 4 biomimetics-10-00553-f004:**
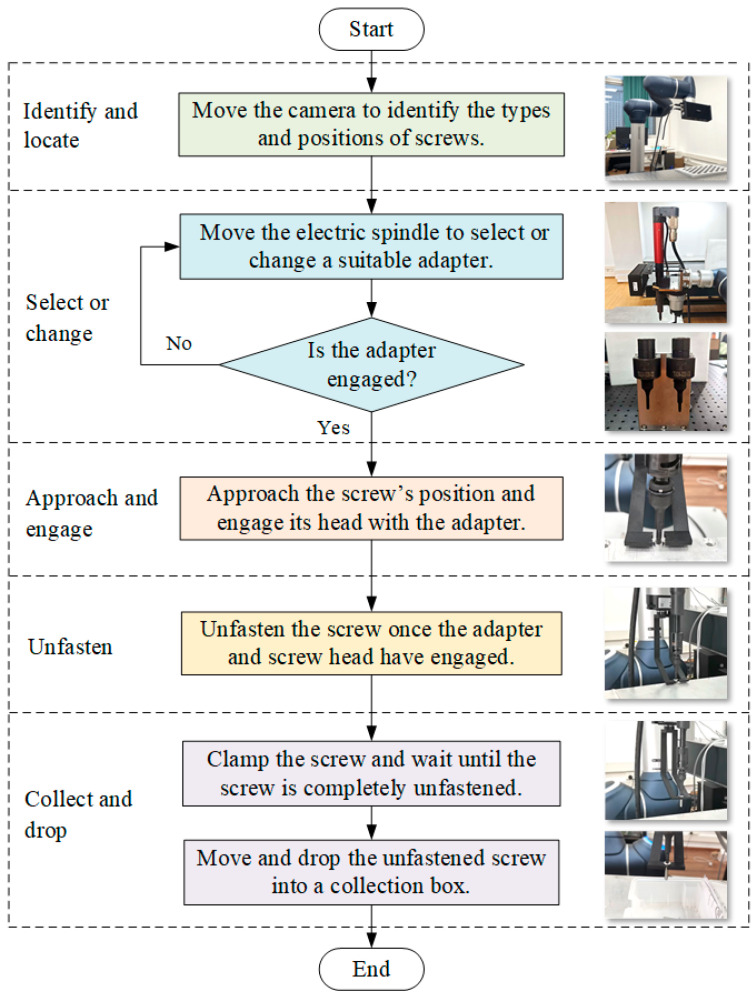
Process of robotic removal and collection of screw.

**Figure 5 biomimetics-10-00553-f005:**
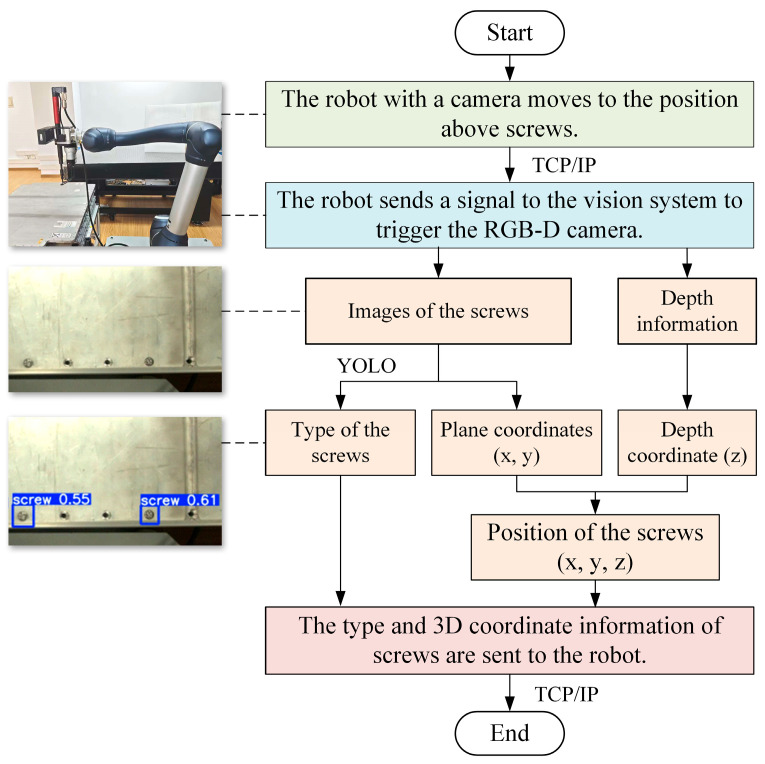
A flowchart of the recognition and localisation of screws based on machine vision.

**Figure 6 biomimetics-10-00553-f006:**
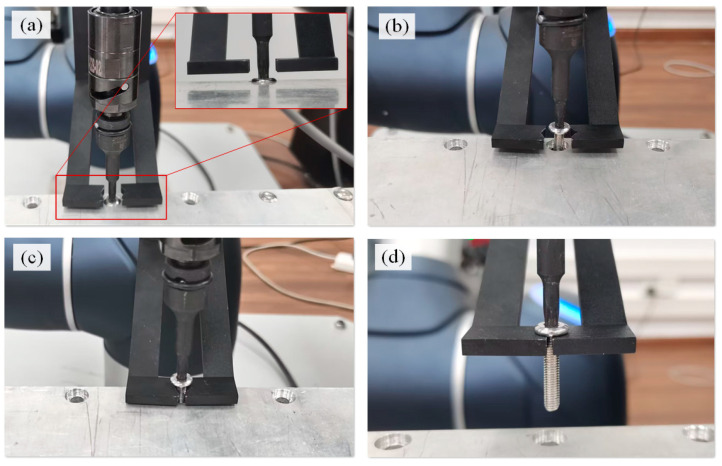
Process of clamping and dropping screws. (**a**) Initial state, (**b**) waiting state, (**c**) clamping state, and (**d**) moving state.

**Figure 7 biomimetics-10-00553-f007:**
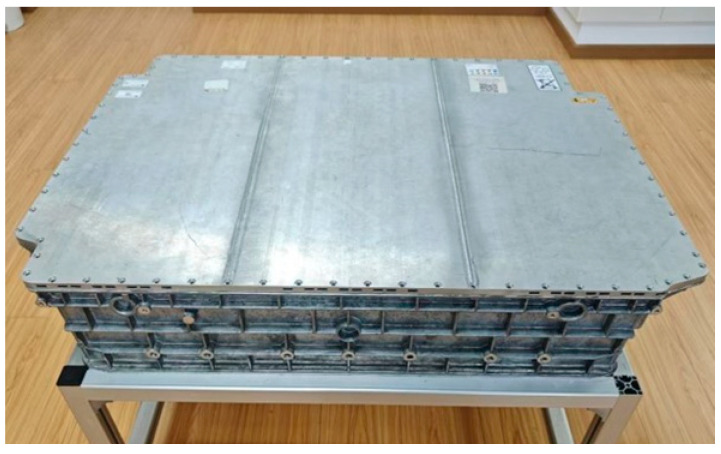
Photo of EoL EV battery.

**Figure 8 biomimetics-10-00553-f008:**
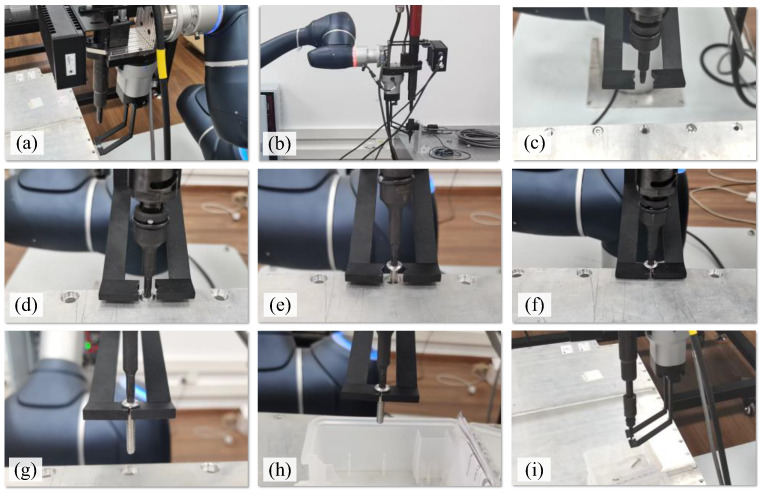
The process of unfastening and collecting screws in an EoL EV battery. (**a**) Image-capturing position. (**b**) Select an adapter. (**c**) Approach the screw. (**d**) Engage the screw head. (**e**) Unfasten the screw. (**f**) Clamp the screw. (**g**) Collect the unfastened screw. (**h**) Move the screw. (**i**) Drop the screw into a collection box.

**Figure 9 biomimetics-10-00553-f009:**
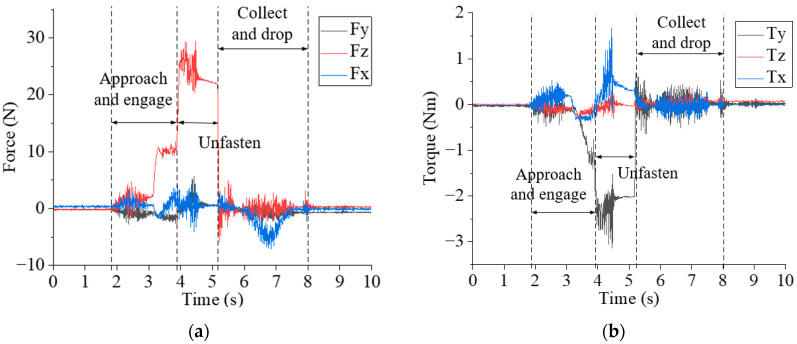
The 6-DOF forces/torques measured by the sensor: (**a**) 3-DOF forces, and (**b**) 3-DOF torques.

**Table 1 biomimetics-10-00553-t001:** Stiffness parameters.

Parameters	Direction	Stages				
		Identify and Locate	Select or Change	Approach and Engage	Unfasten	Collect and Drop
Translational stiffness (N/m)	X	2000	3000	3000	100	2000
	Y	2000	3000	3000	100	2000
	Z	2000	2000	2000	50	2000
Rotational stiffness (Nm/rad)	X	200	100	100	10	200
	Y	200	100	100	10	200
	Z	200	100	100	10	200

**Table 2 biomimetics-10-00553-t002:** The parameters of the electric spindle.

Parameters	Value		
	Select or Change	Approach and Engage	Unfasten
F_Z_ (N)	10	5	15
Rotation speed (rpm)	87.2	43.6	163.8
Min torque (Nm)	2	1	2
Target torque (Nm)	3	2	3
Max torque (Nm)	4	3	4

## Data Availability

The raw data supporting the conclusions of this article will be made available by the authors on request.

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
