# Peer review of "Robotic Removal and Collection of Screws in Collaborative Disassembly of End-of-Life Electric Vehicle Batteries"

_biomimetics, 2025, doi:10.3390/biomimetics10080553_

Round 1
Reviewer 1 Report
Comments and Suggestions for Authors
Dear Authors,
After reviewing the publication, I conclude that it reflects current trends and research directions in the areas of automation and robotization of production processes (in this case, disassembly) and their broadly understood optimization, along with the implementation of sustainable processes.
This work is primarily applied in nature, with little "strictly" scientific content.
I consider the overall layout of the work, along with the division of content and the substantive content of the individual chapters, to be presented at a good level.
However, I suggest:
- in the analysis of the literature in Chapter 2, I propose a systematic review of it and available similar solutions (applications), leading to the identification of areas that have been addressed and presented by the authors in the publication.
- Chapter 5.3 and Figure 8b - the photo does not show the step of replacing the tip (adapter, screw bit) for loosening the screw (this photo differs from Figure 8a in that the head is offset from the battery).
Reviewer 2 Report
Comments and Suggestions for Authors
This is the review of the article “Robotic Removal and Collection of Screws in Collaborative Disassembly of End-of-Life Electric Vehicle Batteries”. The article adopts an intelligent system that automatically detects, removes, and collects the screws from the electric vehicle batteries. Although the topic involves an automatic robot, which doesn’t relate to the journal’s scope, the manuscript is in good structure and involves a good method. However, these are the minor things that need to be considered:
- In this research, the authors developed an expensive system. A 6-DOF robot arm is used to work on the 2D plate surface that only needs a 3-DOF robot arm.
- When the system cannot unfasten a single screw, the battery cannot be disassembled. This is a failed product. The success rate is only 97% which is very low. As can be seen in Figure 7, there are more than 40 screws in the battery. So every 3 batteries, there is a failed product. The defective rate is nearly 30%. That system can not be used in the industry.
- In this system, the authors developed a machine learning model based on the YoLo library that can work normally with many types of cameras, but it can not work well with a specific camera. So, the artificial model didn’t work well. The authors should use a commercial camera with the software inside.
- The authors should compare this system with other systems, such as the traditional method with a fixture system or the hand gesture remote-controlled robotic arm.
- There are many errors in English style and grammar. In particular, there are many typos, such as “space” in line 84,… The authors must check and modify the document in particularly the paragraphs that involve symbols.
